# The Impact of N.I. Vavilov on the Conservation and Use of Plant Genetic Resources in Scandinavia: A Review

**DOI:** 10.3390/plants12010143

**Published:** 2022-12-28

**Authors:** Svein Ø. Solberg, Igor G. Loskutov, Line Breian, Axel Diederichsen

**Affiliations:** 1Department of Agriculture, Faculty of Applied Ecology, Agricultural Sciences and Biotechnology, Inland Norway University of Applied Sciences, P.O. Box 400, 2418 Elverum, Norway; 2N.I. Vavilov Institute of Plant Genetic Resources (VIR), 42-44, B. Morskaya Street, 190000 St. Petersburg, Russia; 3Faculty of Biology, St. Petersburg State University, 7-9, Universitetskaya Emb., 199034 St. Petersburg, Russia; 4Linguistics and Theory of Science, Department of Philosophy, University of Gothenburg, Box 100, 405 30 Gothenburg, Sweden; 5Plant Gene Resources of Canada, Agriculture and Agri-Food Canada, Saskatoon Research Centre, 107 Science Place, Saskatoon, SK S7N 0X2, Canada

**Keywords:** center of origin, crop evolution, diversity, domestication, genebank, plant genetic resources

## Abstract

In this review we examine Nikolai Ivanovich Vavilov’s relationship to Scandinavia and the impact he and his ideas have had on Scandinavia. We trace the historical connections from Vavilov back to 18th century scientists, such as Carl Von Linneaus (Sweden) and 19th century European scientists such as Alphonse de Candolle (Switzerland), Henry de Vilmorin (France), and William Bateson (England). Vavilov has influenced the conservation work in Scandinavia resulting amongst other in the establishment of the Nordic Gene Bank in 1979 and the Svalbard Global Seed Vault which started operating in 2008. Vavilov travelled to Scandinavia in 1921 and in 1931 to give lectures and exchange ideas, especially with the breeders at the Swedish Seed Association (Svalöf) in Scania, Sweden, but also at the Copenhagen University in Denmark. Vavilov did not recognize Scandinavia as part of a center of origin of cultivated plants. It was only after World War II, when P.M. Zhukovsky, a scholar of N.I Vavilov, developed the concept of mega-centers of diversity of cultivated plants, that Scandinavia became part of what he termed the *European-Siberian Region of Diversity*. We list species domesticated in Scandinavia or Northern Europe, and we further discuss concepts related to crop evolution and highlight the great impact Vavilov has had by inspiring scientists across disciplines and over many decades.

## 1. Introduction

Taking a bird’s eye-view, one can see that different regions of the world developed agricultural practices at different paces. Domestication of most farm animals is believed to have started in south-west Asia some 12,000 year ago [1] and domestication of plants in central Asia some 10,000 years ago [2]. Exactly when, why and how this happened is not fully understood. Parallel and independent domestication of plants and animals occurred in different parts of the world. What we know is that some of the oldest domesticated species had features that made them very suitable for the process of domestication. Perhaps there were a few key characteristics that the early innovators looked for. The wild ancestors of domesticated animals like cows, sheep and goats follow a leader, do not easily panic, and are docile in nature. Plants like wheat and pea produce nutritious seeds that can be stored easily. Agriculture practices spread, however, climate similarities/conditions allowed an easier spread along the east–west axis than across geographical latitudes south–north. Therefore, it took 3–4000 years before agriculture that had started in the Near East reached Scandinavia.

Vavilov was interested in the domestication of plants and systematically travelled to collect and document their genetic diversity. Over a 20-year period from 1920 to 1940 he and his colleagues succeeded in building what was then the world’s largest collection of cultivated plants collections with more than 200,000 accessions, collected on 140 national and 40 international expeditions covering 64 countries [3,4]. The book *Five continents* [5] gives an overview of his work, as do the publications by Loskutov [6] and Hummer and Hancock [7].

Vavilov built his ideas on earlier scientists and always acknowledged that. His most recognized research is compiled in the publication *Studies on the origin of cultivated plants* [8]. He wrote this at a time when plant breeding was at a crossroad, with hybridization methods replacing the earlier line selection methods, and the relevance of new genetic material enhance productivity of agriculture became very obvious. In the beginning of the 20th century, most of the plant breeding in Europe was done in small enterprises, and new commercial cultivars were released based on line selections in the existing gene pool. Reading old seed catalogues, it seems there has been a huge diversity, with numerous records for each crop, with different names and line numbers. However, at that time, cultivar naming was flexible and one and the same genotype could be traded with different names or the opposite occurred: the same cultivar name could be used for genetically distinct selections (commercial products) which were indeed very similar. In the 1920s the first official variety tests started in Sweden (as in England), and a more formal seed system developed thereafter [9,10] with the objective to ensure farmers could have confidence the seed lost they purchased.

Vavilov saw the potential in the hybridization methods and he wanted to introduce new, useful and unique donor material for plant breeding. He initiated world-wide collection mission. He brought home a huge diversity of seeds with the aim to support agriculture in the young USSR, a country spanning over many geographical zones. In 1925 he wrote: “Looking at plant breeding for the European and Asian Russia, we have no choice but being geographers” [11].

Vavilov is the pioneer for genebanks for plant genetic resources for food and agriculture in the modern sense and contributed significantly to crop evolution research. Over a twenty-year period, from 1920 to 1940, he headed the VIR genebank [6]. We take the 135th anniversary of N.I. Vavilov as an occasion to consider some of his main ideas, where they came from, but also the impact they have had. We especially focus on his connections to Scandinavia. While Vavilov never considered Scandinavia as a center of crop origin, he nevertheless visited the region to discuss new breeding methods and agriculture in general. Scandinavia was not a hotspot for domestication of very important cultivated plants, but already by the end of the 19th century there were locally and globally well recognized breeding programs located in Sweden and Denmark. In agriculture, Scandinavia was innovative, and at that time the Swedish Seed Association was world leading in breeding innovations and genetics. Vavilov considered Sweden and Netherlands to be the only European countries with a functional plant breeding research after World War I [12].

## 2. Vavilov’s Visits to Scandinavia in 1921 and 1931

Vavilov travelled to Scandinavia once in 1921 and twice in 1931. The trip in 1921 to Scandinavia was part of a longer journey in Europe, and he met with the breeders at the Swedish Seed Association located in Svalöv, in the southern Swedish province of Scania. Vavilov’s predecessor E.R. Regel had already established connections to the Swedish Seed Association on a journey in 1909 [6]. The “modern” plant breeding in Sweden started 35 year before Vavilov’s first visit, with the formation of the aforementioned Swedish Seed Association, which was a farmer-owned national plant breeding institution with a corresponding seed multiplication and marketing structure based on cooperative principles (Svalöf) [13]. 

Vavilov’s second trip to Scandinavia took place in 1931, and then he was invited to several institutions in Copenhagen and Stockholm, as well as in Svalöv and Lund. Hjalmar Nilsson (1856–1925) was the director of the Swedish Seed Association until 1924. At the beginning of the 20th century the varieties that the farmers cultivated consisted of locally adapted populations of different types (landraces or primitive cultivars). For breeders these landraces represented a great potential for line selection, and this was also what Nilsson realized. At the beginning of the 20th century he was known for The Svalöf Method, which was based on individual progenies with a initial selection of single lines followed by subsequent rapid seed multiplications [14] of the most suitable line, a method also applied by Henry de Vilmorin in France at the same time [15]. 

In 1903, the Danish botanist Wilhelm Johannsen had scientifically elaborated the method of developing genetically pure lines in self-pollinated crops by repeated selfing. Johannsen obtained highly homozygous lines that later could be used in crossings [16]. The shift from mass selection or pure-line selection to crossing in cereal breeding came in the first half of the 20th century. At the time of Vavilov’s first visit in 1921, the methods championed by Hjalmar Nilsson at Svalövf were already considered “old school”. Nilsson still worked with what he termed “elementary forms”, which are defined based on morphological botanical characters associated with qualities for cultivation [17]. Nilsson’s background was in botany. He selected morphological lines out from landraces, e.g., in cereals [18]. 

At the Swedish Seed Association Vavilov also met Herman Nilsson-Ehle (1873–1949), most likely both in 1921 and in 1931 [6,19]. Nilsson-Ehle applied the new crossing methods and demonstrated that economically important traits based on multiple genes such as yield also followed Mendel’s laws [20]. He saw the value of using the new crossing technique to combine desirable traits from different genotypes and became the director of the Swedish Seed Association in 1925. He resigned from the position in 1939 [14]. Nilsson-Ehle moved to use induced mutation breeding and made large-scale experiments to create new mutations. One example from the 1930s was to use air balloons to carry seeds up into the stratosphere [21]. There seeds were exposed to radiation and new diversity created due to induced mutation. This method of artificially creating genetic diversity was different form Vavilov’s approach, that looked for genetic diversity in the crops’ center of origin. However, other things Vavilov had in common with the Swedish Seed Association: Vavilov’s institute, similar to Svalöv, had a network of research stations and sub-stations across the country. At a maximum there were 23 such stations in Sweden [22]. Professor Å. Gustafsson at the Swedish Seed Association recalled the conversations with Vavilov during his visit with great warmth. 

At the time of Vavilov’s visits in Sweden there existed two other plant breeding enterprises in Sweden: W. Weibull AB established in 1870, breeding *Brassica* tuber crops, and Sockerbolaget, established in 1907, breeding sugar beet. Vegetable breeding in Sweden started later, with J.E. Ohlsens Enke, L. Daehnfeldts fröhandel and Statens Trädgårdsforsök in the 1930s and1940s. Swedish fruit breeding started at Balsgård in 1941 [22,23]. In Denmark there was plant breeding research at Landbohøjskolen and Copenhagen University, and in enterprises such as A/S Dansk Frøhandel (Trifolium-Silo) (Today DLF Seeds A/S) established in 1900, and the Abed and Pajbjerg Foundations, established in 1903 and 1920, respectively [24].

Vavilov wrote during his visits letters to his colleagues in Leningrad, but did not only report on breeding but also on how farmers put new lands into cultivation and about the many European plant breeders that were on the conferences in Scandinavia [6].

## 3. Vavilov in the Tradition of Earlier European Scientists

Vavilov’s work on crop evolution was grounded on blocks of ideas that were already present. In this section, we discuss how Vavilov’s work was influenced by scientists such as C. Linnaeus, A. de Candolle, H. de Vilmorin, and W. Bateson. 

### 3.1. Vavilov and Linnaeus’s System of Binary Nomenclature 

The Swedish physician and biologist Carolus Linnaeus (1707–1787) established the taxonomy system with plant names based on the genus name and thethe species epithet. This binary nomenclature system and the underlying taxonomic classification very much influenced Vavilov. However, Vavilov did not at all agree with Linnaeus on his view on cultivated species. Linnaeus refused to consider cultivated plants as real species and wrote in harsh words: 

“Cultivated plants are not created, therefore they are not species. All monstrous flowers and plants derive their origin from normal forms. Such monstrosities, variegated, abnormal, multiplied, double, cruciferous, gigantic, wax fat and charm the eye of the beholder with protean variety so long as gardeners perform daily sacrifice to their idol.” From his *Critica Botanica* from 1737, cf. Stearn [25]. 

In the publication “The Linnaean species as a system” [26], Vavilov, despite of Linnaeus’ advice, elaborated on the applications of botanical nomenclature and on the species concept as such in light of the morphological and physiological diversity of cultivated plants: “A Linnaean species is an isolated complex dynamic morpho-physiological system bound in its origin to a certain environment and area.” 

Vavilov researched systematically the enormous variation of many cultivated plant species. The variability of most of the agricultural crops from the temperate zone is documented in numerous volumes of the Cultural Flora of the USSR. The publication of volumes in the series was initiated by Vavilov in 1935 and continued by his associates from the All-Union Institute of Plant Industry (VIR) until the late 1990s [6].

### 3.2. Vavilov and Alphonse de Candolle’s Ideas on Origin of Cultivated Plants

Vavilov was inspired by Alphonse de Candolle (1806–1893), a Swiss-French botanist who worked on crop evolution and geography. In fact, Vavilov (1926) dedicated his great publication “Studies on the origin of cultivated plants” to de Candolle [8]. De Candolle included not only botany in his search for knowledge, but also archaeology, history, and linguistics. In 1883 he published Origine des Plantes Cultivées [27], and he was the first to distinguished between old world and new world crops. 

De Candolle, in contrast to Vavilov, was focusing in his considerations on the species level—and not on particular or distinct entities within a species. However, Vavilov used similar terminology to de Candolle but further developed the ideas. An example is Vavilov’s concept of Centers of origin of cultivated plants or Centers of formation of cultivated plants. These are specific regions of the world with a high morphological diversity, with many distinct forms and with a maximal morphological/genetic diversity. Vavilov developed then the concept of main centers of origin and he particularly listed the crops originated in these centers. He furthermore emphasized mountainous regions or foothills as important origins for crops and distinguished between primary and secondary centers of origin.

### 3.3. Vavilov and Friedrich Alefeld’s Inventory of Cultivated Plants

Friedrich Alefeld (1820–1872) was a German physician and botanist. He was a pioneer in assembling a crop inventory for cultivated plants respecting the genetic diversity within a given species. In his most known publication Agricultural Flora [28], he recognized intraspecific variation of many species, in particular in cereals, and described 1793 botanical varieties. He established the term and taxonomical rank of a “Varietätengruppe”, which was translated to “convarietas” in Latin (convar.), a term that is still in use in agricultural botany. 

### 3.4. Vavilov and Henry de Vilmorin’s Collection of Living Plants

Henry de Vilmorin (1843–1899) was a French botanist and plant breeder. He was the first to establish large living collections for use in plant breeding and is most known for his collections of wheat and sugar beet. Indeed, the collections he built during the second half of the nineteenth century and publication with detailed descriptions were made that inspired Vavilov. 

### 3.5. Vavilov and Bateson’s Ideas of Parallels in Variation

Vavilov developed the idea of parallels in variation, but this was based on earlier works of the English scientist William Bateson (1861–1926), that built on works of Charles Darwin [29]. Bateson, who worked at the John Innes Horticultural Institute at Norwich, England, argued that traits could be predicted based on variation found in other, often related species. One example was the gigantism of certain organs in cultivated species, but there was also a pattern in the phenotypic variation across species [30]. Vavilov regarded parallels in variation as useful knowledge for plant breeding, for example when trying to produce a cultivar with a “new” (so far unknown) seed coat colour in a pulse crop, e.g., pea, that diversity studies had shown to be present in another pulse crop, e.g., lentils. Or when trying to produce cultivars with the absence of a certain alkaloid that so far has not been described in one species but that was found to be absent in another [31]. The breeding of alkaloid free lupins by R. von Sengenbusch was an application of this predictive principle in plant breeding. Vavilov had at that time formulated the Law of homologous series in variation [32], which later has been referred to as a physical co-localization of genetic loci in genetics (synteny) [33]. Parallels can be found in the physical co-localization of genes on the chromosomes. Bateson, one year before his death, visited Vavilov in Leningrad to see the experiments and discuss the theory. Later, Vavilov developed the idea further into what he termed Ecological passport [34]. He divided the Old World into 19 areas, where in each area he could see a pattern in the local varieties, having similar, general characteristics. He established such ecological passports for cereals, pulses, as well as for oil- and fibre flax, and linked this to parallelisms in evolution. The work came late in his career and was published in 1940 [35]. Most of Africa and tropical Asia were not included in this work nor was the New World (America and Oceania). 

### 3.6. Vavilov and Interdisciplinarity

Vavilov was one of several scientists who pioneered an effort to create an interdisciplinary framework for combining natural sciences and the humanities to gain more comprehensive and integrative insights in the past. In a recent book, Elena Aronova shows how Vavilov’s research built a bridge between biologists in the Soviet Union and historians in France with the *Annales school of historiography* [36]. Cross-disciplinary efforts such as these were ahead of their time and point at the challenges arising from disciplinary divisions.

### 3.7. Vavilov’s Acknowledgement

Vavilov fell victim to Stalinism and died in prison in Saratov 1943. In his works he acknowledged the famous scientists mentioned above, and made reference to many others. Moreover, he acknowledged the support and outstanding contributions of a large team of colleagues. Several of them continued his work and secured the collections under very difficult circumstances during World War 2 and the decades that followed. Among these colleagues we find E.I. Barulina, K.A. Flaksberger, P.M. Zhukovsky, E.A. Stoletova, A.P. Popova, N.R. Ivanov, E.N. Sinskaya, S.M. Bukasov, V.S. Lekhnovich, N.V. Zinger, and E.V. Wulff and many others [37].

## 4. From Vavilov Onwards

In the following section, we show examples of the impact of Vavilovs work up until today. We start with a bibliometric analysis. Thereafter, we examine how Vavilov impacted developments in crop evolution research and conservation in the mid- and late 20th century. In this context, the establishment of international and national genebanks are central, and we draw lines from Vavilov to the Nordic Genebank and to the Global Seed Vault at Svalbard. 

### 4.1. Bibliometric Analysis

Web of Science is the world’s largest scientific database covering all disciplines and all types of scientific literature. A search on Vavilov as topic (TS = Vavilov), and including only the Web of Science Categories “Plant science”, “Genetics Heredity”, and “Agronomy”, resulted in 173 records. Plotting the publications by time shows that Vavilov is still relevant. However, there are a few clear peaks (Figure 1). The first peak is in 1987, which matches the 100 years anniversary of Vavilov. This second peak is not so easy to explain, but it may reflect an increasing awareness of conserving biodiversity. We should be careful to over-interpretate such bibliometric data, but a closer look shows that many of the captured papers deal with centers of origin and confirm the delimitation of the gene centers, e.g., the American center [38].

### 4.2. Crop Evolution and Centers of Origin

Evgeniya Nikolaevna Sinskaya (1889–1965) continued Vavilov’s work on crop evolution and centers of origin. She added an African center of origin to Vavilov¨s centers [39]. Peter M. Zhukovsky (1888–1975) became the director of the Vavilov institute for a ten-year period. He developed the concept of “Mega-gene-centers” but also “Endemic micro-gene-centers”. He created world maps listing cultivated plant species belonging to the mentioned ccenters. With A. Zeven from the Netherlands, Zhukovsky published the book “Dictionary of cultivated plants and their centers of diversity” [34].

In Germany, Elisabeth Schiemann (1881–1972) was inspired to conduct work on diversity of cultivated plants from secondary centers of origin. She also worked on crop wild relatives [40,41]. 

In the USA, Harry Harlan (1882–1944) worked on barley and applied many of the principles Vavilov had established; they communicated with each other. His son Jack Harlan and Marc de Wet introduced the gene pool concept, where wild progenitors were classified into primary, secondary, and tertiary gene pools according to their potential utilization in breeding [42]. Harry Harlan, and his son Jack Harlan (1917–1998), framed what they termed Agricultural crop centers and non-centers [43]. They were convinced that the true centers were a few regions of special importance for domestication of plants, like the Fertile Crescent, East Asia and Meso America. Non-centers, on the other hand, were the larger parts of the rest of the world to where crops were spread and additional diversity developed [44]. 

Inspired by Vavilov; Purugganan and Fuller [45] applied archaeological information and proposed 24 areas around the world that could be grouped into 13 centers of domestication. Other scholars that Vavilov influenced are C.O. Sauer, I.H. Burkill, C.D. Darlington, A.I. Kuptsov, R. Porteres, and J.G. Hawkes [46]. Nigel Maxted from Birmingham University has worked on crop wild relatives and their geography [47,48]. In a recent investigation, Maxted and Vincent [49] investigate to which extent Vavilov’s centers are congruent with areas of high diversity of crop wild relatives, the CWR hotspots, and found a close geographic coincidence but not a full match. However, Vavilov’s center had a better match than Zhukovsky’s mega-centers [34], Harlan’s centers and non-centers [43], and Purugganan and Fuller’s centers [45]. 

### 4.3. Conservation of Crop Diversity

The impact Vavilov had on conservation of crop diversity is huge. His work inspired many others to collect, conserve and research the diversity of cultivated plants. 

One person that certainly was inspired by Vavilov was the Austrian/Australian geneticist Sir Otto Herzberg Frankel (1900–1998). Frankel was very concerned about the loss of local landraces. The green revolution replaced many of these varieties [50]. Frankel, together with the Scottish breeder Erna Bennett (1925–2012), who at that time worked for the Food and Agriculture Organization of the United Nations (FAO), called for conservation actions to avoid severe genetic erosions [51,52,53]. Their work influenced the outcome of the United Nations Conference on the Human Environment that took place in Stockholm in 1972. The result was a global programme on the conservation of genetic resources organized by The International Board for Plant Genetic Resources (IBPGR). 

### 4.4. The Impact of Vavilov in Scandinavia

With his work on crop evolution and conservation of crop diversity, Vavilov had a great impact on the Scandinavian countries. On the question of taxonomy, however the Vavilovian School did not fully match the Swedish School. Vavilov and many of his scholars applied the taxonomic principles established by Linnaeus and expanded them to structure and describe the intraspecific diversity of crop gene pools. This approach has been disputed in particular by the western school. For example, Mac Kay did not acknowledge the hierarchical taxonomic classifications that Vavilov and his colleagues developed in wheat, with botanical variety classifications based on morphology [54]. Mac Kay argued that these classifications did not reflect the evolutionary relationships and had limited relevance for breeding, simply because most modern wheat cultivars all fall into the same botanical variety: Triticum aestivum L. var. lutescens (Alef.) Mansf.

Many national genebanks were established from 1970 onwards, influenced by contemporary ideas that were inspired by Vavilovs work. In 1979, the Nordic Gene Bank (NGB) was established. The first director, Ebbe Kjellqvist, had worked for FAO, and had definitely read Vavilov, as he worked with institutes in Turkey, Syria, Iraq, Iran, Afghanistan and Pakistan collecting genetic resources in this recognized Near East center of origin [55]. He attended the 1972 Stockholm Conference when establishing a joint Nordic genebank was on the agenda [56]. Prior to this, on a European plant breeding conference in 1966, regional collaborations on conservation was also discussed, and Scandinavia was mentioned as one region amongst others. Ebbe Kjellqvist and Stig Blixt became the primary instigators for such a regional genebank establishment in the Nordic countries [57]. While the VIR collection has material from all over the world, the Nordic Gene Bank (now part of Nordic Genetic Resource Center) has as a strategy to only conserve material of Nordic origin. In Sweden and Denmark there were however already large university collections with, e.g., barley, wheat, and pea accessions, consisting of accessions had been collected from all over the world, including in the centers of origin, on collection missions organized with Scandinavian researchers. Birger Kajanus at Weibullsholm (Sweden) established the pea collection in the 1920s, and with colleagues at Weibullsholm he built what became the largest pea collection in the world, with 3427 accessions at a maximum [9]. The accessions included around 200 described genes. Most accessions were characterized with 45 traits including quantitative traits as yield, protein content and flowering time [58]. When breeding of peas declined the collection was donated to the Nordic Gene Bank and the John Innes Centre (England) and most of the accessions are still maintained and available for distribution. 

The Svalbard Global Seed Vault was opened in 2008 in a partnership between the Norwegian Government, the Nordic Genetic Resource Center, and the Global Crop Diversity Trust. This happened as a result of a long and still ongoing process of collaboration and trust building among collection holders and among countries [59] The Global Seed Vault at Svalbard today houses 1.165.041 accessions of seed germplasm from 5.947 species from all around the world [60]. The seed vault is an emergency back-up facility for the many genebanks that carry out the footwork in maintaining living collection through regeneration, quality controls, viability tests, characterization and evaluations and distribution of seed germplasm to users. 

The Crop Trust is an organisation created by FAO and Bioversity International to primarily support the ex situ conservation and use of crop diversity. Thus, we see a direct link between Vavilov’s ideas and the establishment of the Svalbard Global Seed Vault. Indeed, Vavilov must have inspired the many persons behind the establishment of the seed vault: like Carry Fowler, but also the first coordinators of the seed vault Ola Westengen and the current coordinator Åsmund Asdal.

## 5. Scandinavia in the Context of Crop Evolution

### 5.1. Scandinavia in the European-Siberian Region of Diversity of Cultivated Plants

When the domestication was initiated in Asia, the entire Scandinavia was still covered with snow and ice, and thereafter land was covered with rocks, bare soils or water for many centuries. Vavilov did not indicate Scandinavia as part of a center of origin of cultivated plants of global significance. Grains, pulses, and vegetable germplasm were brought here from south or east, via trade routes, e.g., from the Mediterranean. Agriculture reached southern Scandinavia around 6000 years ago [61]. Since then, various crops have been adapted to the short, cool growing season with very long days. Over time, locally adapted and very specific crop diversity has developed in Scandinavia in many crops, e.g., for example in Brassica crops, rye, and oats. Still, Scandinavia, expect for Brassica napus ssp. napobrassica, has not been referred to as a secondary center of diversity for these species. 

Darlington [62] described Europe as a region of origin, for example for some important fruit and berries as well as forage crops. The mega centers that Zeven and Zhukovsky [34] suggested, also included parts of Scandinavia as the European-Siberian Region of Diversity and listed around 190 species including wild relatives belonging to this region. Among them we also find various vegetables, herbs, berries (Table 1). These plants have been collected but also cultivated [63,64]. Some of them were multifunctional. Angelica (*Angelica archangelica* L.) was used in spirituous, leaves for salads, and young stems as sweets. The plant was cultivated in home gardens in Iceland and Norway. Roots were brought onboard when the wikings travelled. They are rich in vitamin C, which was important on long journeys with little access to fresh food. In Norway, there is a special local variety ‘Vossakvann’ with filled petioles [65]. This is most likely a mutation that has been maintained by local farmers. It is now conserved and distributed by the Norwegian Seed Savers (https://kvann.no/, accessed on 1 October 2022). Although this specific genotype is conserved, a review on a broader set of traditional vegetables from Scandinavia showed that there are gaps in the conservation [66]. The Scandinavian region is especially rich in berries, and berries have both been collected and cultivated and are essential sources of vitamin C and other micronutrients. Gooseberry (*Ribes uva-crispa* L.), redcurrant (*Ribes sativum Syme*), black currant (*Ribes nigrum* L.), and raspberry (*Rubus idaeus* L.), but also cloudberry (*Rubus chamaemorus* L.) are examples of such berries. These also grow wild and the region with genotypes with extremely winter-hardiness and other useful traits. Germplasm of fruits and berries are kept in field genebanks where wild material is usually not present. 

### 5.2. The ‘Ecological Passport’ of Scandinavia

Vavilov linked the northern parts of Scandinavia and most of Finland to territory number 19 (Northern Agricultural Territory), an area that also included Northern European USSR and Siberia [34]. Crop plants in this territory shared general characteristics such as: (a) low heat requirements, (b) cold resistance, and (c) medium size seeds. Examples named by Vavilov included very early types of barley and rye.

The southwestern part of Scandinavia and southern Finland were linked to the territory number 17 (West-European Agricultural Territory). Crop plants from this area have general characteristics as: (a) being hydrophytic, (b) having thick and stiff stems, (c) being tall, (d) having large and broad leaves, (e) relatively early maturity; and for cereals (f) having large and dense spikes with medium-sized or large grains.

## 6. Conclusions

We have seen that Vavilov built his great ideas on centers of diversity on a cross-disciplinary understanding of various European scholars. We know he extensively travelled and collected crop diversity and built a global collection of a wide specter of species. He visited Scandinavia in 1921 and 1931 and exchanged ideas with the scientists there. One example is the network of research stations and sub-stations across the country that was present in Sweden at that time and also what Vavilov had established back home. The impact of N.I. Vavilov is visible in many ways. In Scandinavia, we see a direct link to the establishment of the Nordic Gene Bank in 1979 and the widespread collection and conservation work that followed. Vavilov inspired the persons behind this initiative, such as Ebbe Kjellqvist Stig Blixt and others. We also see a connection from Vavilov to the many scientists that have been involved in collection missions, conservation and use of plant genetic resources worldwide, and finally the persons behind the establishment of the global back-up facility at the Svalbard Global Seed Vault. Even today, 135 years after Vavilov’s birth, his works inspire young generation of scientists in genebanks and beyond.

## Figures and Tables

**Figure 1 plants-12-00143-f001:**
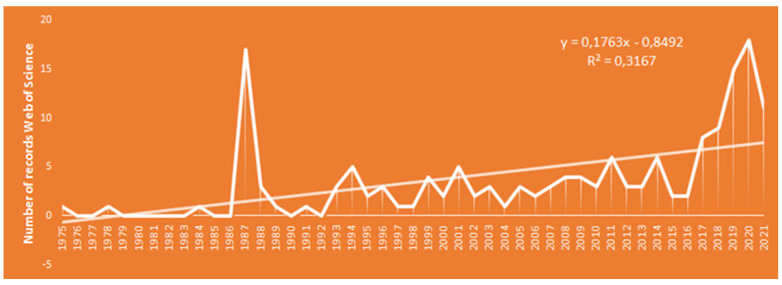
Number of records on Web of Science using a search on Vavilov as topic (TS = Vavilov), and with a refining including only the Web of Science categories “Plant science”, “Genetics Heredity”, and “Agronomy”. Results are provided from 1975 onwards with a trendline.

**Table 1 plants-12-00143-t001:** Examples of crops from the European-Siberian region and with Scandinavia as part of this region. Species are extracted from the original source [34] but checked with current distribution maps using GBIF [67]. The comments are our own or modified from the original source.

Species	Common Name	Comment
Forage legumes		
*Anthyllis vulneraria* L.	Spring vetch	Formerly cultivated as pasture mix
*Astragalus cider* L.	Milk vetch	Formerly cultivated as pasture mix
*Astragalus falcatus* Lam.	Sicklepod milk vetch	Formerly cultivated as pasture mix
*Coronilla varia* L.	Crown vetch	Formerly cultivated as fodder crop
*Lathyrus sylvestris* L.	Flat pea	Formerly cultivated as fodder crop
*Medicago falcata* L.	Yellow lucerne	Wild relative to lucerne, E Europe
*Trifolium hybridum* L.	Alsike clover	Possibly first cultivated in Sweden
*Trifolium repens* L.	White clover	Cultivation started in N Italy and Holland
*Trifolium pratense* L.	Red clover	Cultivation started in UK and Holland
Forage grasses		
*Alopecurus pratensis* L.	Meadow foxtail	Formerly cultivated
*Agropyron cristatum* (L.) Gaertn.	Crested wheatgrass	Formerly cultivated
*Agrostis canina* L.	Velvet bentgrass	Formerly cultivated in the Netherlands
*Arrhenatherum elatius* (L.) P. Beauv. Ex J & C Presl.	Tall meadow oat grass	A valuable pasture grass and CWR
*Bromus erectus* Huds.	Erect brome	Formerly cultivated, E and C Europe
*Elymus caninus* L.	Bearded wheatgrass	Formerly cultivated, E Europe
*Festuca ovina* L.	Sheep’s fescue	Cultivated
*Festuca pratensis* Huds.	Meadow fescue	Common forage crop
*Festuca rubra* L.	Red fescue	Common forage crop
*Phleum pratense* L.	Timothy	Common forage crop
*Dactylis glomerata* L.	Orchard grass	Common forage crop
*Poa pratensis* L.	Meadow grass	Common forage crop
*Poa palustris* L.	Marsh meadow grass	Arctic Europe, varieties developed
Vegetables and herbs		
*Humulus lupulus* L.	Hops	Common spice plant
*Angelica archangelica* L.	Angelica	Formerly cultivation in Scandinavia
*Carum carvi* L.	Caraway	Common spice plant
*Atriplex hortensis* L.	Garden orache	Formerly cultivated
*Allium scorodoprasum* L.	Sand leek	Formerly cultivated in the USSR
*Rumex acetosa* L.	Garden sorrel	Formerly cultivated in N Europe
*Beta vulgaris* L.	Beet root	Wild along the coast
*Brassica rapa* L.	Turnip	Root and leafy types, Finland
*Brassica napus* ssp. napobrassica (L.) Rchb.	Swede	Europe is a secondary gene center
Fruit and berries		
*Ribes nigrum* L.	Black currant	Very winter-hardy types in Scandinavia
*Ribes uva-crispa* L.	Gooseberry	Cultivated in temperate region
*Ribes sativum* Syme	Redcurrant	Cultivated in temperate region
*Rubus arcticus* L.	Arctic raspberry	Rich in vitamin C, used in breeding
*Rubus chamaemorus* L.	Cloudberry	Partly domesticated, used in breeding
*Rubus idaeus* L.	Raspberry	Widely cultivated
*Rubus saxatilis* L.	Stone bramble	Resistant to rust and other diseases
*Fragaria* spp.	Strawberries	Includes wild strawberry (*F. vesca* L.)

## Data Availability

Not applicable.

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
