# Peer review of "The Impact of N.I. Vavilov on the Conservation and Use of Plant Genetic Resources in Scandinavia: A Review"

_plants, 2022, doi:10.3390/plants12010143_

Round 1
Reviewer 1 Report
N.I. Vavilov was a fouder of "genetic resources science", his work was essential for breeding in agriculture, for genetics, agro-botany, and most of other plant sciences. Progress in breeding for resistance, tolerance, quality, content of nutrients and other compounds was done generally on the basis of Vavilov s work. Therefore, I consider the review of Vavilov´s global impact useful and important and here it was devoted Scandinavian region, that is also interesting. Scandinavia is not a hotspot for PGR, but for species occuring there and for the work done in the region the review is exhausting. It covers Vavilov´s early travels to the region and his scientific contacts including early breeding. The article covers also his connection to important old European scientists - Linneus, de Candole etc.. Analyzing of WOS records reflects rather conferences to Vavilov´s anniversaries than increase of scientific impact and is not reliable. It was very useful also to mention Vavilov´s succeeders, Sinskaya, Zhukovskij, Harlan... Impact of Vavilov´s work for Scandinavia is finally highligted in opening of the Global Seed Vault in Svalbard, it can be agreed. Scandinavia belongs to Euro-Siberian Region of diversity, that is very important. Listing of examples of crop species is good but the group of fruits and berries in underestimated, I think this is the main input of the region to the world herritage. The authors should come with much more examples and pay attention to them in the text. Vice versa listing of common forage grasses is not so important, there are many interesting boreal grass species that are completely omitted, similarly some boreal legumes. The authors should include those more interesting species to make the table "more attractive". The title, abstract, key words and chapters seems comprehensive, including references. Conclusions highlight only Global Vault, that is not enough. English must be made more clear and fluent, some parts are like google translated from Russian (Russian construction of sentence, many embedded sentences). I propose to reformulate some complicated and not understandable sentences. Corrections are shown or proposed in the text. I recommend to accept the text after proposed completing the text and corrections.
Author Response
Reviewer 1
N.I. Vavilov was a founder of "genetic resources science", his work was essential for breeding in agriculture, for genetics, agro-botany, and most of other plant sciences. Progress in breeding for resistance, tolerance, quality, content of nutrients and other compounds was done generally on the basis of Vavilov s work. Therefore, I consider the review of Vavilov´s global impact useful and important and here it was devoted Scandinavian region, that is also interesting. Scandinavia is not a hotspot for PGR, but for species occurring there and for the work done in the region the review is exhausting. It covers Vavilov´s early travels to the region and his scientific contacts including early breeding. The article covers also his connection to important old European scientists - Linneus, de Candole etc.
Thanks, no comment
Analyzing of WOS records reflects rather conferences to Vavilov´s anniversaries than increase of scientific impact and is not reliable.
Thank you for this comment and we modified the interpretation of this bibliometrics. We still kept the figure to illustrate the 100 years peak that the interest for Vavilov is still present in scientific community. The changes we made are highlighted in the revised version.
It was very useful also to mention Vavilov´s succeeders, Sinskaya, Zhukovskij, Harlan...
Thanks, no comment
Impact of Vavilov´s work for Scandinavia is finally highlighted in opening of the Global Seed Vault in Svalbard, it can be agreed.
Thanks, no comment
Scandinavia belongs to Euro-Siberian Region of diversity, that is very important. Listing of examples of crop species is good but the group of fruits and berries in underestimated, I think this is the main input of the region to the world heritage. The authors should come with much more examples and pay attention to them in the text. Vice versa listing of common forage grasses is not so important, there are many interesting boreal grass species that are completely omitted, similarly some boreal legumes. The authors should include those more interesting species to make the table "more attractive
Thanks for this comment! We clearly see your point and added more forages, vegetables, and berries, and we added running text highlighting some details of these species, what they have traditionally been used for and sone on the conservation status. We clearly see that we can expand this section but leave this to another paper focusing on genetic resources from the region. Table 1 has been extensively expanded with examples. The changes we made are highlighted in the revised version.
The title, abstract, key words and chapters seems comprehensive, including references. Conclusions highlight only Global Vault, that is not enough.
Thank you for this comment, and we have expanded the conclusion to reflect more of the issues we raised. The changes we made are highlighted in the revised version.
English must be made more clear and fluent, some parts are like google translated from Russian (Russian construction of sentence, many embedded sentences).
Thank you for this comment. You might be right with a couple of sentences that have been translated from Russian, but 995 was actually written by a native Norwegian speaking person. We might have some of the similar style than the Russian language. Nevertheless, we have had the entire manuscript checked and reread by a English speaking person, and changed a lot in the manuscript
I propose to reformulate some complicated and not understandable sentences. Corrections are shown or proposed in the text. I recommend to accept the text after proposed completing the text and corrections.
We have reformulated complicated sentences as advised. We are sorry for the errors that were made and have corrected all these sentences. All changes are highlighted in the revised manuscript.
Reviewer 2 Report
For a Russian person, the role of N.I. Vavilov in the development of genetics, breeding and biology in general is extremely important.
The huge collections he collected are still in demand. The development of new research methods allows us to take a fresh look at this valuable material.
The review is devoted to Vavilov's connectionsб collaborations with Scandinavia, his role in the study of biodiversity there, the creation of genebanks, and the development of the ideas of Scandinavian biologists.
The review is read with great interest, and can be useful not only to researchers, but also to teachers of universities and colleges when preparing lectures on genetics, biodiversity, and breeding.
Author Response
For a Russian person, the role of N.I. Vavilov in the development of genetics, breeding and biology in general is extremely important. The huge collections he collected are still in demand. The development of new research methods allows us to take a fresh look at this valuable material. The review is devoted to Vavilov's connections б collaborations with Scandinavia, his role in the study of biodiversity there, the creation of genebanks, and the development of the ideas of Scandinavian biologists. The review is read with great interest, and can be useful not only to researchers, but also to teachers of universities and colleges when preparing lectures on genetics, biodiversity, and breeding.
Thank you for your positive feedback and that you used your time to review this manuscript. We have made some changes based on the feedback from a second reviewer and hope that this has not changed your opinion.
Round 2
Reviewer 1 Report
The text of contribution was greatly improved in the new version. However, many comments and recommendations were not addressed. There are still parts that would need reformulation in order the text be more fluent and understandable for readers. Especially formulations derived from Russian language should be expressed by a better English. I do not have additional comments, I still insist on the previous, the authors should go through my recommendations in the correction notes of the text and try to improve the style.